# Multiple Administration Routes, Including Intramuscular Injection, of Oncolytic Tanapoxvirus Variants Significantly Regress Human Melanoma Xenografts in BALB/c Nude Mice Reconstituted with Splenocytes from Normal BALB/c Donors

**DOI:** 10.3390/genes14081533

**Published:** 2023-07-27

**Authors:** Michael L. Monaco, Omer A. Idris, Grace A. Filpi, Steven L. Kohler, Scott D. Haller, Jeffery E. Burr, Robert Eversole, Karim Essani

**Affiliations:** 1Laboratory of Virology, Department of Biological Sciences, Western Michigan University, Kalamazoo, MI 49008, USA; michael.l.monaco@wmich.edu (M.L.M.); omer.a.idris@wmich.edu (O.A.I.); grace.a.filpi@wmich.edu (G.A.F.); steve.kohler@wmich.edu (S.L.K.); rob.eversole@wmich.edu (R.E.); 2Charles River Laboratories, Mattawan, MI 49071, USA; scott.haller@crl.com; 3PharmOptima LLC, Portage, MI 49065, USA; jeburr@pharmoptima.com

**Keywords:** melanoma, immuno-oncolytic viruses, xenograft, adoptive transfer, immune reconstitution, tanapoxvirus, immunovirotherapy, virotherapy

## Abstract

Human melanoma is the most aggressive form of skin cancer and is responsible for the most deaths of all skin cancers. Localized tumors, and those which have limited spread, have 5-year survival rates of over 90%, with those numbers steadily rising over the past decade. However, metastatic melanomas have a sharp decrease in 5-year survival rates and are still an area of need for new, successful therapies. Immuno-oncolytic viruses (OVs) have emerged as a promising class of immunovirotherapy that can potentially address this disease. The Food and Drug Administration in the United States has approved one oncolytic herpes simplex virus expressing granulocyte-macrophage colony-stimulating factor (Talimogene Laherparepvec) for the treatment of metastatic melanoma, and others could soon follow for this and other cancers. In previous studies, Tanapoxvirus (TPV) recombinants expressing mouse interleukin-2 (mIL-2) and another expressing bacterial flagellin from *Salmonella typhimurium* (FliC) have demonstrated anti-tumor efficacy in nude mouse models. TPV replicates only in humans and monkeys, including tumor cells, which makes the use of syngeneic tumor models impossible for the study of this OV in a standard immunocompetent system. In this study, TPV/Δ66R/mIL-2 and TPV/Δ2L/Δ66R/FliC were tested for their ability to treat human melanoma xenografts (SK-MEL3) in a BALB/c nude mouse model reconstituted with splenocytes from genetically compatible, normal BALB/c donor mice. Two SK-MEL3 tumors were transplanted into each flank of BALB/c nude mice, and the larger tumor was treated intratumorally (IT) with virus or mock injection. In one set of animals, mice received adoptive transfers of splenocytes from BALB/c mice on day 4 to reconstitute their immune systems and allow for adaptive immune responses to occur in a xenograft model. Direct IT injection of TPV/Δ66R/mIL-2 led to significantly greater rates of tumor regression compared to reconstituted control (RC) mice in the primary and distant tumor sites, whereas TPV/Δ2L/Δ66R/FliC treatment led to significantly greater rates of tumor regression in distant tumor sites only. A second experiment used TPV/Δ66R/mIL-2 to test whether TPV could be administered intravenously (IV), intramuscularly (IM), or both routes simultaneously to exert similar anti-tumor effects in an indirectly treated tumor. A single SK-MEL3 tumor was transplanted into one flank of BALB/c nude mice and was treated either into the tail vein, the nearest rear leg to the tumor, or both. All mice then received adoptive transfers of splenocytes in the same way as previously described on day 4 and received an additional TPV treatment on day 14. The results demonstrated that TPV/Δ66R/mIL-2 treatment IV or IM had significantly greater rates of tumor regression than RC-treated mice but failed to exert this effect when both routes were used simultaneously. Data obtained through these experiments support the continued development of Tanapoxvirus for the treatment of human melanoma and using immune reconstitution to create intact adaptive immunity in a xenograft context, which can allow other tropism-limited OVs to be studied against human cancers.

## 1. Introduction

Melanoma is the most aggressive and lethal form of skin cancer, even though it accounts for only about 1% of all skin cancer diagnoses. In 2023, it is estimated that around 98,000 people will be diagnosed with cutaneous melanoma in the United States, which will lead to around 8000 deaths [1]. Skin cancer is the most diagnosed cancer type in the United States, and although mortality rates have steadily declined over the past decade or so for all skin cancers, including melanoma, the quantity of diagnoses globally necessitates new and effective treatments. Advanced, metastatic melanoma, however, is still very dangerous, with survival rates lowering from >99% when localized to 32% when spread to distant sites [1].

The current standard of care for metastatic melanoma includes a range of immunotherapies, including immune checkpoint inhibitors (anti-CTLA-4 and anti-PD-1 antibodies), interleukin-2 (IL-2) and tumor necrosis factor (TNF) infusions, chemotherapy and radiotherapy in some cases, targeted therapies such as BRAF (v-raf murine sarcoma viral oncogene homolog B1) and MEK (mitogen-activated protein kinase kinase) inhibitors, angiogenesis inhibitors (anti-VEGF antibodies), and even oncolytic virotherapy from FDA-approved Talimogene Laherparepvec (Imlygic) [2]. Surgery is utilized where applicable and is the main curative treatment for localized tumors.

In this study, an oncolytic virus (OV) being developed for the potential treatment of metastatic melanoma, Tanapoxvirus (TPV), was tested for efficacy in a new xenograft mouse model. TPV is a poxvirus (dsDNA), distantly related to the vaccinia virus, which uniquely causes a very mild, self-limiting infection with no reported transmission between humans. Previous experiments in our lab have shown that TPV recombinants, including one that expresses mouse IL-2 with the viral thymidine kinase gene (*66R*) deleted, were effective at inducing significant tumor regression in an outbred immune-deficient nude mouse model [3,4]. The previous tumor model did not include the testing of another TPV recombinant, which has an additional deletion of viral tumor necrosis factor binding protein (*2L*) and the insertion of bacterial flagellin from *S. typhimurium* (FliC). It was included in the presented experiments due to its prior efficacy in other tumor models tested in our lab [5,6] and that many cancers upregulate toll-like receptor 5 (TLR5). The activation of TLR5 by bacterial flagellin can induce anti-tumor effects in cancers [7]. In melanoma, TLR5 is not as well studied as TLR2 or TLR4, but evidence suggests that many cell lines express TLR5 to varying degrees [8], and therefore, innate and adaptive immunity could be activated by TPV/Δ2L/Δ66R/FliC infections in melanoma cells.

However, due to TPV’s host range limitations to humans and monkeys (including cancerous tissues), standard syngeneic tumor model testing is not possible. Therefore, to test TPV as a prospective OV in an immune-competent system, one of two possibilities exist prior to direct human trials: a tumor model in monkeys or the creation of a new model system. Since there are no commercially available tumor cell lines from monkeys, making a new model system involving xenografts is the only option. Our lab has tried different methods for the creation of such a model in the past [6], and we believe the current iteration of this model can be applied to other tropism-limited OV candidates.

Altering routes of administration is also a key obstacle in many clinical applications of OV therapies. Generally, OVs have the highest rates of success if they can be administered directly into a tumor to exert direct lytic effects on tumor cells and express therapeutic transgenes to stimulate immune cell recruitment and activation in tumors where the microenvironment may be actively suppressing immune cell activity. However, not only are many tumors not directly accessible for intratumoral (IT) injection, surgical interventions to allow for direct injection may bring too great a risk to a patient’s life and undue expense if multiple doses are required [9]. Therefore, the clinical preference is for intravenous (IV) delivery, which also allows greater compatibility with potential adjuvant chemotherapies or immunotherapies that are delivered IV. Many different approaches are being used to combat obstacles presented to OV therapeutic success brought by exposing the virus to immune cells in the vasculature, such as infection of immune cells as carriers, nanoparticle viral encapsulation, or tumor retargeting on the viral surface to enhance binding affinity to tumor cells. In other cases, injected titers are merely increased, knowing many virus particles will be victims of phagocytosis and will not arrive at the tumor site [10,11]. Interestingly, intramuscular (IM) administration of OVs has not been used as the sole route of administration for the treatment of cancers, even though it is the standard way of delivering vaccines.

Here, we describe the evaluation of TPV recombinant viruses expressing mIL-2 and FliC against SK-MEL3 human melanoma xenografts in BALB/c nude mice (CAnN.Cg-*Foxn1^nu^*/Crl) reconstituted via adoptive transfer of splenocytes from normal BALB/c donors. In the first experiment, after the induction of two human melanoma tumors on each flank of the BALB/c nude mice, TPV treatment was injected IT into one tumor, leaving the other untreated. Four days later, splenocytes from genetically identical, normal BALB/c mice were injected intraperitoneally (IP) into mice groups that were to receive adoptive transfers and tumor measurements were conducted every 2 days for a total of 40. In a similar experiment, we also evaluated different routes of administration for the TPV recombinant expressing mIL-2 to include IV and IM injections. Only a single SK-MEL3 tumor was induced in each BALB/c nude mouse, and in this case, TPV/Δ66R/mIL-2 was the sole virus tested. TPV was injected on day 0, mice were reconstituted with splenocytes on day 4, and then a second TPV injection was given on day 14, with tumor measurements occurring in the same intervals as before for a total of 40 days. We report herein that multiple TPV recombinants are capable of inducing significant tumor regression of human melanoma xenografts in the primary site and in a distant tumor site, as well as TPV/Δ66R/mIL-2 inducing significant tumor regression of human melanoma xenografts when administered IV or IM.

## 2. Materials and Methods

### 2.1. Cells, Viruses, and Reagents

Owl monkey kidney (OMK) cells and human triple-negative cancer (MDA-MB-231) cells were purchased from American Type Culture Collection (ATCC, Rockville, MD, USA) and cultivated at 37 °C with 5% CO_2_. OMK cells were grown in Eagle’s minimum essential medium (EMEM), supplemented with 100 U/mL penicillin, 100 µg/mL streptomycin, 30 mL/L 7.5% NaHCO_3_, and 10% (*v*/*v*) fetal bovine serum (FBS). SK-MEL3 cells were grown in McCoy’s 5A medium, supplemented with the same components as with the EMEM and 15% FBS. For recombinant virus replication, OMK cells were infected as described previously. Expression of the IL-2 and FliC transgenes was previously confirmed [3,5]. TPV/Δ66R/mIL-2 was previously tested for efficacy in an immunodeficient (IDt) animal model bearing human melanoma xenografts in our lab [12].

### 2.2. Virus Amplification

Both viruses used were previously amplified in OMK cells and concentrated 100× using ultracentrifugation (45Ti rotor at 186,000× *g* for 90 min). The concentrated viruses were then titrated in 6 well dishes containing OMK cell monolayers as described previously [13]. Viruses were diluted in sterile phosphate buffered saline (PBS-A) to 5 × 10^6^ plaque-forming units (PFU) per 100 µL. Viruses were stored at −80 °C and, when needed for virotherapy, thawed on ice at 4 °C, sonicated for 8–10 s for uniform suspension, and kept on ice during injections.

### 2.3. Animals

Balb/C athymic nude mice (CAnN.Cg-*Foxn1^nu^*/Crl) and Balb/C normal mice (BALB/cAnNCrl) were purchased between 4 and 5 weeks of age from Charles River and acclimated for 1 week following arrival. All animals were housed, and subsequent treatments were carried out following protocols approved by Western Michigan University’s Institutional Animal Care and Use Committee (IACUC number 19-06-04).

### 2.4. Virotherapy of Human Melanoma Xenografts in BALB/c Nude Mice

Tumor xenografts were induced by injection of 5 × 10^6^ SK-MEL3 cells, mixed 1:1 (*v*/*v*) with Matrigel (Corning Life Sciences, USA), and suspended in 100 µL of sterile Dulbecco’s PBS (DPBS). Both flanks of female BALB/c nude mice (~6 weeks) were injected subcutaneously. Cell viability was tested both prior to injection and post injection with 0.4% (*w*/*v*) trypan blue dissolved in PBS-A to ensure that >90% of the cell population was and remained viable through the procedure. Once tumors became visible, volumes were measured with digital Vernier calipers using the formula ((length × width × height) × (π/6)) in mm^3^. When 1 of the 2 transplanted tumors reached the range of 120 mm^3^–180 mm^3^, mice were randomly assigned into 1 of 8 possible treatment groups, half being IDt and the others receiving adoptive transfer of splenocytes. In virotherapy-treated groups, a single dose of 5 × 10^6^ PFU/100 µL (suspended in sterile DPBS) TPV/Δ66R/mIL-2, TPV/Δ2L/Δ66R/FliC, or wild-type (wt) TPV was delivered IT into the bigger of the 2 tumors, with the smaller remaining untreated. Tumors from either of the control groups (mock—MC or reconstituted—RC) were injected IT with 100 µL of vehicle into the bigger of the 2 tumors, with the smaller remaining untreated. This represented day 0 of treatment. Group average tumor volumes on day 0, ± 1 standard error, are presented in Appendix A. Tumor volumes for all tumors were then measured independently every other day for a period of 40 total days. Following humane sacrifice, the remaining tumor tissue and blood samples were taken, with tumors fixed in 10% formalin for 48–72 h, then transferred to 60% ethanol for long-term storage at 4 °C and blood samples placed at 37 °C for 30 min to coagulate, refrigerated overnight at 4 °C, then centrifuged at 12,000 rpm for 10 min at 4 °C to collect serum (Eppendorf 5403 R).

### 2.5. Adoptive Transfer of Splenocytes from BALB/c Donor Mice

On day 4 post therapy, mice assigned to ICt groups were reconstituted with whole splenic cells from genetically identical BALB/c normal donor mice. Donor mice were sacrificed via cervical dislocation, and the whole spleen was collected aseptically. The spleen was placed in ice-cold RPMI-1640 + 10% FBS, placed onto a metal sieve, and pressed through using a plastic spatula until the organ disintegrated and individual cells were collected into the medium. The cell mixture was then centrifuged at 1100 rpm for 5 min at 4 °C. Following centrifugation, cells were resuspended in ice-cold DPBS and centrifuged again under the same conditions, repeated twice. Cells were then tested for >90% viability as described before and diluted to ~3 × 10^6^ cells/100 µL. Cells were kept on ice and were injected intraperitoneally (IP) into each mouse.

### 2.6. Statistical Analyses

We used linear mixed models to analyze the effects of the treatments on tumor growth curves [14,15]. Tumor volume on day 0 varied substantially among mice, so the response variable in all analyses was the percent of initial tumor volume (i.e., tumor volume on day 0). The percentage of initial volume was log-transformed to improve model fit.

Tumor growth curves were modeled using second-order orthogonal polynomial regression. We used orthogonal polynomials to eliminate collinearity between linear (*Day*) and quadratic (*Day*^2^) time terms. We initially built four nested models. Model 1 contained only orthogonal linear and quadratic time terms. Model 2 included orthogonal linear and quadratic time terms and virus treatments without any interactions between virus treatments and time. Model 3 was identical to Model 2, but it included interactions between the virus treatments and the linear time term. Finally, Model 4 was the same as Model 3, but it included interactions between the virus treatments and both time terms. All four models included random effects for each mouse (intercepts and both the linear and quadratic time terms). The models were compared using likelihood ratio tests. All analyses were carried out in R version 4.2.2 (R Core Team 2023) using the *lme4* package (version 1.1-33). *p*-values were obtained using *lmerTest* (version 3.1-3), and comparisons of treatments means were obtained using *phia* (version 0.2-1). A separate analysis was performed for each of the following groups: the injected tumor in reconstituted mice, the injected tumor in deficient mice, the non-injected tumor in reconstituted mice, and non-injected tumors in the deficient mice. In all four cases, Model 3 provided the best fit to the data and is the basis for all results reported below. We used residual plots and normal probability plots to examine whether Model 3 satisfied assumptions of the linear mixed effects model. These plots suggested that assumptions of the statistical model were not violated in any of the four cases.

To determine if the effects of virus treatment on tumor growth depended on immune status, we performed separate analyses, which omitted the mock control and reconstitution control treatments. Model 1 contained only orthogonal linear and quadratic time terms. Model 2 included orthogonal linear and quadratic time terms, virus treatments, immune status (competent, deficient), and their interaction, but no interactions with time. Model 3 was identical to Model 2, but it included interactions between the linear time term and both the virus treatments and immune status. Finally, Model 4 was the same as Model 3, but it included interactions between the quadratic time term and both the virus treatments and immune status. Separate analyses were performed for the injected tumor and the noninjected tumor. In both cases, Model 4 provided the best fit to the data and is the basis for the results reported in Section 3.3. As before, we used residual plots and normal probability plots to examine whether Model 4 satisfied assumptions of the linear mixed effects model. These plots suggested that the assumptions of the statistical model were not violated in either case.

## 3. Results

BALB/c nude mice were injected subcutaneously on both flanks with 5 × 10^6^ SK-MEL3 human melanoma cells, which grew until one tumor reached a volume between 120 and 180 mm^3^. At that point, the larger of the two tumors was injected intratumorally (IT) with 5 × 10^6^ PFU/100 μL of either TPV or sterile buffer (PBS-A) to serve as a mock injection. Tumor volumes were measured every 2 days, with multiple independent measurements for a total of 40 days. Both the directly injected and non-injected contralateral tumors were measured during this time period. The results of this experiment are presented in Figure 1. All TPV treatments were significantly more effective at tumor regression than mock treatment when injected IT in BALB/c nude mice (Figure 1a), consistent with previous experiments in outbred nude mice [3]. In non-injected contralateral tumors, TPV recombinants were also able to exert anti-tumor effects in a statistically significant manner when compared to mock treatment (Figure 1b, *p* = 0.01011). Analysis of the log percent of initial tumor volume at the experimental midpoint (day 21) showed that the mean slopes of the growth curves for recombinant TPV-treated mice were significantly smaller than the mean of wtTPV and mock-treated tumors (*p* = 0.009553).

### 3.1. BALB/c Nude Mice Bearing SK-MEL3 Xenografts Are Significantly Regressed in Both Injected and Non-Injected Tumors When Treated with TPV Recombinants in a Splenocyte Reconstituted Model

Similar to the IDt model, BALB/c nude mice were injected twice subcutaneously with 5 × 10^6^ SK-MEL3 cells, with the tumors allowed to grow to the same 120–180 mm^3^ initial volume threshold prior to treatment. The major difference in this experiment was that all treated mice were injected IP with 3 × 10^6^ splenocytes from genetically identical, normal BALB/c donor mice on day 4 following IT treatment into the larger tumor on day 0. All analyses compared TPV treatments against reconstituted control (RC) mice, where they received mock injections of sterile buffer but were given splenocytes to reconstitute their immune systems. The experiment was carried out for 40 days, with measurements of tumor volume taking place once every 2 days for both directly injected and non-injected contralateral tumors. For the directly injected tumors, TPV/Δ66R/mIL-2 and wtTPV exerted significant rates of tumor regression compared to the RC group (*p* = 0.01550 and *p* = 0.04921, respectively), whereas TPV/Δ2L/Δ66R/FliC was not as effective (*p* = 0.43224). These results are shown in Figure 2a. Interestingly, for the contralateral, non-injected tumors, the effects of the immunomodulatory transgenes carried by the TPV recombinants tested appeared to be playing a significant role. The mean log percent of initial tumor volume for both TPV recombinant treatments compared to wtTPV and RC treatments was significantly less and demonstrated tumor regression (*p* = 5.78 × 10^−6^). When compared independently, TPV/Δ2L/Δ66R/FliC treatment led to significantly greater regression than RC (*p* = 0.005195), and TPV/Δ66R/mIL-2 treatment led to greater regressions compared to RC, though not significant (*p* = 0.087323). Although there was no difference between TPV recombinant efficacy in the non-injected tumors (*p* = 0.27794), wtTPV was significantly worse in exerting anti-tumor effects on the contralateral tumors than the RC treatment alone (*p* = 0.03992). These results are shown in Figure 2b.

### 3.2. Endpoint Analysis at Day 38 of Injected and Non-Injected Tumors, Irrespective of Immune Status, Shows That TPV/Δ66R/mIL-2 and TPV/Δ2L/Δ66R/FliC Exerted Significant Anti-Tumor Effects Compared to Control Treatment for Directly Injected Tumors, and Nearly Significantly Greater Anti-Tumor Effects Than wtTPV in Non-Injected Tumors

After analysis of tumor data in both directly injected and non-injected tumors for both immune status groups, an endpoint analysis was performed to compare static volume differences between treatments on day 38. Data points from all groups were analyzed and grouped by treatment type, regardless of immune status, to determine if any treatment was demonstrating significantly different average tumor volumes by the end of the experiment. Due to these comparisons being non-planned in nature, all data were adjusted using the Holm method to maintain data integrity; *p* values were multiplied by the number of total comparisons (6). For all injected tumors, when comparing the mean of MC and RC group volumes for both IDt and ICt mice on day 38 to the means of TPV treatments, TPV/Δ2L/Δ66R/FliC and TPV/Δ66R/mIL-2 both demonstrated significantly reduced tumor volumes at the experimental endpoint (*p* = 0.01635 and *p* = 0.01159, respectively). Mean control tumor volumes compared to wtTPV treatment was not significant (*p* = 0.26675). None of the TPV treatments were different from one another for directly injected tumors. The results are presented in Table 1. For non-injected tumors, no comparisons were significantly different, though treatment with both TPV recombinants was nearly significantly less than wtTPV treatment (*p* = 0.05279 for TPV/Δ2L/Δ66R/FliC and *p* = 0.05149 for TPV/Δ66R/mIL-2). However, the mean log-transformed tumor volume for wtTPV treatments in the non-injected tumors was significantly greater than the mean volume of TPV/Δ2L/Δ66R/FliC and TPV/Δ66R/mIL-2 treatments combined (*p* = 0.004536).

### 3.3. Immune Status of TPV Treated BALB/c Nude Mice Impacts Treatment Efficacy in Non-Injected Tumors, but Not in Directly Injected Ones

Two separate analyses using a factorial ANOVA were conducted on collected tumor volume data for directly injected and non-injected tumors to determine whether the immune status of the treated animals had an impact on TPV-mediated anti-tumor efficacy. For the injected tumors, tumor growth did not differ between TPV treatments (*p* = 0.2773) or immune status (*p* = 0.2679), and there was no interaction between TPV treatment and immune status (*p* = 0.4378). For non-injected tumors, there was also no interaction between TPV treatment and immune status; however, there were significant effects of both TPV treatment and immune status on tumor volume in the case of the non-injected tumors. When comparing TPV treatments in non-injected tumors, the mean log percent of initial tumor volumes for wtTPV-treated mice were significantly greater than both TPV recombinants over the course of the experiment (*p* = 6.172 × 10^−5^), with neither TPV/Δ2L/Δ66R/FliC nor TPV/Δ66R/mIL-2 being significantly different from one another (Figure 3b). When compared at midpoint (day = 21), the slopes of the growth curves for wtTPV-treated animals were significantly greater compared to the mean of both TPV recombinants (*p* = 2.482 × 10^−5^). Though the effects of both immune status and TPV treatment individually were significant in non-injected tumors, where the mean percent of initial tumor volume was significantly lower (18%) in splenocyte-reconstituted mice than in immune-deficient mice (Figure 3a), the interaction between immune status and TPV treatment was not significant (*p* = 0.188018).

### 3.4. Two Doses of TPV/Δ66R/mIL-2 Intravenously or Intramuscularly Leads to Significantly Regressed Tumor Volumes Compared to RC Treatment

We aimed to determine whether TPV could demonstrate any significant efficacy in the treatment of melanoma tumors in this reconstituted BALB/c nude mouse model if we changed the route of administration from IT to intravenous (IV), intramuscular (IM), or both simultaneously (IV + IM). For this experiment, we chose to only use the TPV/Δ66R/mIL-2 recombinant as it had demonstrated efficacy IT for the reconstituted animals, and IV-delivered interleukin-2 has been FDA approved since 1998 for the treatment of stage IV metastatic melanoma [16]. In this experiment, BALB/c nude mice were injected subcutaneously with a single SK-MEL3 xenograft, which was allowed to grow to 100 mm^3^ in volume, and then the first administration of TPV/Δ66R/mIL-2 took place with 1 × 10^6^ PFU/50 µL representing day 0. Four animals were randomly assigned to be treated either IV via the tail vein, IM via the rear leg nearest to the tumor, or both IV and IM, where half of the volume was delivered into each of the previously mentioned locations simultaneously. On day 4, each mouse received adoptive transfers IP as was done previously, using 3 × 10^6^ splenocytes from normal BALB/c donor mice. Then, on day 14, a second administration of TPV/Δ66R/mIL-2 took place in the same location(s) as before for each group. Tumor volumes were measured once every 2 days for a total of 40 days, using Vernier calipers as described previously. Comparisons were made to the non-injected tumors of the RC group, where the group mean initial tumor volume was just greater than 100 mm^3^ on day 0.

Analysis of the log percent of initial tumor volume demonstrated that both IV and IM administrations of TPV/Δ66R/mIL-2 regressed tumor volumes significantly faster than RC treatment (*p* = 1.57 × 10^−5^ and *p* = 0.00278, respectively), whereas simultaneous administration of TPV/Δ66R/mIL-2 IV and IM did not induce the same effect (*p* = 0.27486). When comparing the slope of tumor growth trajectories (interaction between treatment and Day at the midpoint of the experiment (day 21)), IV administration of TPV/Δ66R/mIL-2 was significantly more negative than RC-treated mice (*p* = 0.03563) with IM administration of TPV/Δ66R/mIL-2 being less negative than RC treatment, but not significantly (*p* = 0.08241). When comparing whether the shape of the growth curve at day 21 was the same in all treatments (interaction between treatment and Day^2^), both IV and IM administration of TPV/Δ66R/mIL-2 demonstrated a significantly more convex growth curve shape (i.e., more hump-shaped) than RC treatment (*p* = 0.008985 and *p* = 0.010054, respectively). There was no difference between the shape of the growth curve for IV + IM administration of TPV/Δ66R/mIL-2 and RC treatment (*p* = 0.101026). The results are presented in Figure 4 below.

## 4. Discussion

In previous studies from our lab, we demonstrated that TPV recombinants could significantly regress human melanoma xenografts in an outbred nude mouse model [3,4]. To further evaluate TPV as a potential OV, tumor models in immune-competent animals are the logical next step. Due to TPV’s natural host tropism being limited to humans and monkeys, there is no available syngeneic model that can be used to further the investigation of TPV as an oncolytic. The creation of a new model closely resembling a syngeneic mouse was necessary. A nude mouse strain inbred from BALB/c mice allowed for the adoptive transfer of splenocytes from genetically compatible donor mice. This effectively reconstitutes the nude mice with functional, mature T cells and other immune cells that allow for intact immune responses against protein-based antigens. Even though an obvious feature of a xenograft model that is reconstituted with spleen cells is the eventual regression of the tumor based on transplant rejection mechanisms (presuming T cell-dependent immune responses become functional), the timing of adoptive transfer and the number of infused cells has been adjusted to allow for a window of time where anti-tumor responses led by an OV can be measured and analyzed for statistical significance. Indirect evidence of the T cell-dependent immune response being intact in these mice following immune reconstitution is the tumor volume decreasing in RC-treated mice, which do not occur in MC-treated immune-deficient animals. The only difference between these two groups is the adoptive transfer of splenocytes in the case of the RC animals, and their tumors begin to decrease around day 22 in both directly injected and non-injected tumors, where both tumors continue growing in MC mice.

In previous investigations of TPV-mediated anti-tumor efficacy against human melanoma, recombinants expressing mIL-2 and a recombinant with the viral 15L gene deleted (neuregulin-like protein) both demonstrated transgene expression and significant reductions in tumor volume compared to mock injected control animals [3]. In those studies, the recombinant TPV/Δ2L/Δ66R/FliC was not tested. The FliC recombinant has shown efficacy against colorectal cancer xenografts in nude mice, with confirmed gene expression [5] and in a model similar to the one described here [6]. The flagellin protein from *S. typhimurium* is a potent activator of the innate immune response through binding of toll-like receptor 5 (TLR5), and many cancer cells will express TLR5 even though the normal tissue does not [17]. Although TLR5 is not a prognostic marker in melanoma, various studies have shown that TLR5 agonists can lead to synergistic anti-tumor properties with ICIs and other therapies in cancer types, including melanoma [17,18]. Therefore, we believe an investigation of TPV/Δ2L/Δ66R/FliC against human melanoma should be included with TPV/Δ66R/mIL-2 in this in vivo study.

In the first experiment, BALB/c nude mice were injected with two SK-MEL3 tumors, with one being injected directly and the other left without treatment. The primary idea was to simulate having a large metastatic lesion and determine whether an IT injection with TPV recombinants could lead to tumor regression in the primary as well as a distant site. For these BALB/c nude mice, the average time from inoculation of SK-MEL3 cells to treatments on day 0 was 5 weeks. This experimental design was applied in both immune-deficient BALB/c nude mice and in reconstituted mice. In immune-deficient mice, all TPV treatments, including wtTPV, led to significant tumor regression when compared to mock-treated control animals for tumors directly injected with the virus. In tumors contralateral to the primary site, efficacy was to a lesser degree, though there was still a significant reduction in the rate of tumor growth for TPV/Δ2L/Δ66R/FliC- and TPV/Δ66R/mIL-2-treated mice when compared with wtTPV and mock control treatments. This highlights the importance of TPV’s stimulation of the immune system via inserted transgenes, as both TLR5 and mIL-2 signaling activate cells of the innate immune response, though the full potential of immune activation is compromised. Therefore, anti-tumor efficacy is going to be a function of direct tumor cell lysis by TPV and innate immune system activation in the immune-deficient nude mouse model. Since the virus was injected IT into only one tumor, TPV would need to leave the initial tumor site via vasculature created by both tumors to reach the contralateral site. Likely, small numbers of particles made it to the secondary site to induce direct infection of the tumor cells there, where primary anti-tumor responses to lessen growth rates may have been initiated by activated innate immune cells.

When tested in splenocyte-reconstituted BALB/c nude mice, the effects of TPV’s anti-tumor activity were also robust. Mice injected IT with TPV/Δ66R/mIL-2 and wtTPV had significantly greater rates of tumor regression than the RC-treated mice, whereas TPV/Δ2L/Δ66R/FliC treatment induced a more minimal and non-significant effect on the rate of tumor regression in the primary tumor site. We believe that wtTPV outperformed TPV/Δ2L/Δ66R/FliC in this circumstance due to the increased viral replication possible by wtTPV in the early stages of treatment. This can be seen in Figure 2, where the growth trajectory in the primary tumor sites for wtTPV-treated mice was a flat–convex shape compared to the more arched convex curve in the TPV/Δ2L/Δ66R/FliC treatment. The reduction in initial tumor growth for wtTPV likely led to the overall significant difference in mean log percent initial tumor volume compared to RC, even though TPV/Δ2L/Δ66R/FliC-treated tumors were smaller than wtTPV-treated tumors by day 40. After the experimental midpoint (day 21), the rate of tumor regression in TPV/Δ2L/Δ66R/FliC-treated mice was increasing marginally every measurement period. Had the experimental timeline been allowed to proceed until tumor resolution, this treatment likely would have induced significantly faster regression than RC for directly injected tumors in this model.

Yet, when analyzing the anti-tumor effects exerted on the non-injected, distant tumor site, both TPV/Δ2L/Δ66R/FliC and TPV/Δ66R/mIL-2 treatment had significantly greater rates of tumor regression compared to RC-treated animals. Even more telling of the impact the immunomodulatory genetic insertions in each virus had on treatment, was how wtTPV-treated mice had significantly greater tumor volumes than RC treatment for non-injected tumors. It was the only group where the tumor volume was increasing over time, despite these animals having the ability to reject the tumors once reconstituted. This further supports the idea that the recombinant TPVs are superior candidates for development as OVs than wild-type.

The endpoint analysis of directly injected and non-injected tumors also corroborates the conclusion that both TPV/Δ66R/mIL-2 and TPV/Δ2L/Δ66R/FliC are superior OVs to wtTPV. The data presented in Table 1 combined and averaged tumor volumes in both immune-deficient and splenocyte-reconstituted animals by treatment type. Statistical analysis showed that by the end of the experiment, regardless of whether the mice had been reconstituted or not, recombinant TPV treatment was exerting significant effects on tumor volume compared to control treatment in directly injected tumors. wtTPV was not demonstrating these same anti-tumor effects. In the non-injected tumors, after Holm adjustment for non-planned comparisons, there were no significant differences between any TPV treatment and control treatment. However, the mean of the wtTPV treatments compared to the mean of both recombinant TPVs combined was significantly greater (*p* = 0.004536). This indicates that treatment after 38 days with wtTPV was significantly less effective at treating distant tumor sites than recombinant TPVs. This is once again most likely the effect of immune cell stimulation from inserted FliC and mIL-2, which leads to anti-tumor effects in these non-injected sites.

To determine whether the efficacy of the TPV recombinants against the melanoma xenografts was dependent on the immune status of the treated mice, factorial ANOVA was used. Separate analyses were performed for the directly injected and non-injected tumors. In both analyses, no evidence of an interactive effect between the immune status of the mice and the TPV treatments was found. For the directly injected tumors, the *p*-values from the factorial ANOVA for effects of treatment, immune status, and interactions between treatment and immune status were all greater than 0.25. This made direct comparisons unnecessary for the directly injected tumor sites. The comparisons of differential treatment efficacy in the non-injected tumors were clearer. It was determined that the effects of treatment and immune status were significant for non-injected tumors, yet the interaction between treatments and immune status was not. The log percent of initial tumor volume for wtTPV-treated mice was significantly greater than both TPV/Δ66R/mIL-2 and TPV/Δ2L/Δ66R/FliC treatments, irrespective of immune status. The slopes of the growth curve at day 21 were also significantly greater in wtTPV-treated mice than the mean slopes of both TPV recombinants. Without recruitment of the immune system toward tumor sites where initial virus titer would be non-existent, wtTPV does not exert anti-tumor effects in this model. The mean log percent of initial tumor volume for all TPV treatments in splenocyte-reconstituted mice was significantly less (18%) than in the immune-deficient, TPV-treated animals. This demonstrates the effect of reconstitution on overall treatment performance and why having immune competency in animal models is critical to a more realistic analysis of an OV’s efficacy.

In a real cancer treatment scenario, it is less likely that a tumor can be directly accessible to IT treatment, particularly in metastatic cancer, even though melanoma tumors are generally more accessible than other solid tumor types. This is why we also investigated whether TPV/Δ66R/mIL-2 could be administered IV (via the tail vein), which is standard for chemotherapeutics and other molecular therapies, IM (into the rear flank nearest to the tumor), which is standard for vaccinations, or both routes simultaneously. The experimental design was slightly modified from the first tumor model to only have a single tumor location since all tumors would be indirectly treated, and to include a second dose of virus on day 14. The initial tumor volumes were treated slightly sooner than in the IT experiment (100 mm^3^ threshold versus 120–180 mm^3^), as this was near the approximate average tumor volume in the RC mice for the non-treated tumors on day 0. Finally, the dose of TPV/Δ66R/mIL-2 was much less than the IT dose (1 × 10^6^ PFU/50 µL) to account for adding a second dose of TPV, which we have not tested in any previous models. In this experiment, TPV/Δ66R/mIL-2 demonstrated significantly faster rates of tumor regression for both IV- and IM-treated mice when compared to mock treatment. This further supports the conclusions from the comparisons of non-injected tumors in the splenocyte-reconstituted IT model; TPV is capable of inducing significant anti-tumor effects at a distant site, regardless of direct or indirect treatment intervention. The combination of IV and IM treatment together was not very effective in contrast to either alone. This observation could potentially have been due to the delivered titer of TPV being too little in either location (5 × 10^5^ PFU/25 µL was administered into each location on the same day) to exert any anti-tumor effects at the tumor site. Since both IV and IM administrations of TPV/Δ66R/mIL-2 alone demonstrated equivalent tumor regression in this model, it is unclear at this time whether simultaneous administration has an interference effect or if the minimal titer required to induce anti-tumor effects was not achieved by this approach.

IM administration has been tested as an immune priming technique for subsequent IV injections of oncolytic maraba virus in a metastatic lung model of murine melanoma, where activation of CD8^+^ T-cells facilitated significantly increased survival rates [19]. A future investigation could test whether this strategy would be more effective than administration of IV and IM simultaneously for TPV in this model. IM delivery would potentially add the benefit of increased ease of administration for the OV. If efficacy levels were statistically equivalent in a clinical setting, IM administration allows for easier use in situations where IV delivery is difficult for either the patient or the facilities the patient is being treated in are underequipped.

There were a number of potential limitations associated with our study. First, as this study was designed primarily to test for TPV-mediated anti-tumor efficacy in this specialized animal model, the assessment of tumor growth modeling was one of the key parameters around which our conclusions were drawn. However, there is a lack of accompanying pathological, tumor-infiltrating lymphocyte, and cytokine analyses, which would have further strengthened the idea that the TPV recombinants tested are capable of inducing potent anti-tumor responses whether injected directly into a tumor or indirectly administered. Our lab has shown in previous experiments that monocytes and other lymphocytes have been recruited by transgenes expressed through the infection of treated tumor cells in other nude mice models with melanoma and triple-negative breast cancer (TNBC) [3,12] and in currently unpublished data for a BALB/c nude mouse model similar to the one described, bearing TNBC tumors. Though not demonstrated directly, we are confident that similar cellular responses observed in previous studies would apply here as well, based on tumor volume data. Secondly, this study does not address to what degree direct infection of tumor cells by TPV contributes to observed anti-tumor efficacy in the systemic treatment of melanoma xenografts. Future studies should determine viral titers within systemically treated tumors progressively throughout a treatment period in this model. Understanding the potential differences in viral depletion capability that nude mice have without mature T cell populations and with them restored following adoptive transfer can help determine the ultimate translatability of systemic TPV treatment in vivo to immune-competent human patients. Finally, demonstrations of neutralizing antibody production in nude mice following adoptive transfer are not shown in the presented experiments. This would provide direct evidence that these mice have a complete restoration of immune functionality. These experiments are currently ongoing in our laboratory.

## 5. Conclusions

In this study, we investigated the efficacy of two TPV recombinant viruses, TPV/Δ66R/mIL-2 and TPV/Δ2L/Δ66R/FliC, in a new, splenocyte-reconstituted BALB/c nude mouse model against SK-MEL3 human melanoma xenografts. Both viruses demonstrated significant tumor volume regression in different circumstances when injected intratumorally, with TPV/Δ66R/mIL-2 being a slightly stronger overall candidate than TPV/Δ2L/Δ66R/FliC in the melanoma xenograft models presented herein. Additionally, TPV/Δ66R/mIL-2 also demonstrated the ability to significantly regress melanoma xenografts compared to control in our model when injected either intravenously or intramuscularly. The model presented in this manuscript could also become a new study system for in vivo investigations of other tropism-limited OV candidates that have an available xenograft model in immune-deficient mice but have not been able to use a model with a functional T cell-dependent immune response. Taken together, we believe there is strong evidence to support the further development of TPV/Δ66R/mIL-2 for the treatment of human melanoma.

## Figures and Tables

**Figure 1 genes-14-01533-f001:**
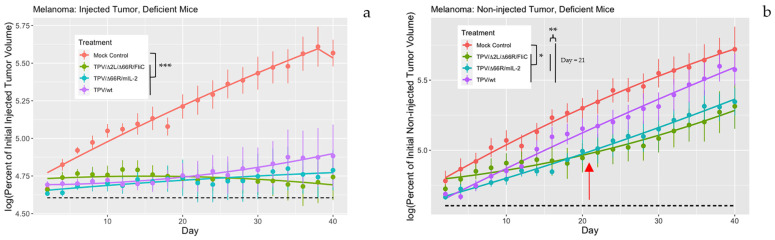
TPV treatment of human melanoma xenografts in BALB/c nude mice. Mice were injected subcutaneously with 5 × 10^6^ SK-MEL3 cells on both flanks and allowed to grow until one tumor reached 120–180 mm^3^ in volume. Once one tumor reached this threshold, it was injected directly with either 5 × 10^6^ PFU/100 μL of TPV or sterile buffer for a mock injection. Both directly injected and non-injected tumor volumes were measured every 2 days for a period of 40 days total. Measurements were taken with Vernier calipers with at least two independent measurements per day. Volumes were calculated using the formula: ((length × width × height) × (π/6)). All tumor volumes were expressed as a percent of volume on day 0, then log transformed, with the *y*-axis representing the logarithm of percent initial tumor volume, where values above the dotted line are greater than the log of 100 (4.60517). The *x*-axis represents days post treatment. In both (**a**,**b**), the red line represents mock treatment, green represents TPVΔ2L/Δ66R/FliC, blue represents TPV/Δ66R/mIL-2, and purple represents wild-type TPV. In (**a**), all TPV treatments significantly regressed tumor volumes when directly injected into the tumor compared to mock control treatment (*p* = 1.823 × 10^−7^ ***). There was no difference between any TPV treatments for the directly injected tumors. In (**b**), the mean log percent of initial tumor volume for all TPV treatments averaged together was significantly regressed in contralateral, non-injected tumors compared to mock control (*p* = 0.01011 *). Rate of tumor regression was also compared at the midpoint of the experiment (red arrow) for non-injected tumor treatments as a slope of the growth curve. The mean slopes for mock control and wtTPV were significantly greater than the mean of TPV/Δ2L/Δ66R/FliC and TPV/Δ66R/mIL-2 at day 21 (*p* = 0.009553 **). There was no difference in slope between mock control and wtTPV or between TPV/Δ2L/Δ66R/FliC and TPV/Δ66R/mIL-2 at day 21. * *p* < 0.05, ** *p* < 0.005, *** *p* < 0.0005, *n* = 5–6 per group.

**Figure 2 genes-14-01533-f002:**
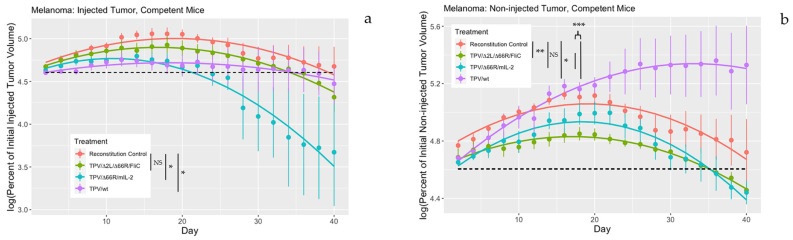
TPV treatment of SK-MEL3 xenografts in BALB/c nude mice reconstituted with splenocytes from normal BALB/c donors. Mice were injected subcutaneously with 5 × 10^6^ SK-MEL3 cells on both flanks and allowed to grow until one tumor reached 120–180 mm^3^ in volume. Once one tumor reached this threshold, it was injected directly with either 5 × 10^6^ PFU/100 μL of TPV or sterile buffer for a mock injection (day 0). On day 4, all mice were injected intraperitoneally with 3 × 10^6^ splenocytes from normal BALB/c donor mice. Both directly injected and non-injected tumor volumes were measured every 2 days for a period of 40 days total. Measurements were taken with Vernier calipers with at least 2 independent measurements per day. Volumes were calculated using the formula: ((length × width × height) × (π/6)). All tumor volumes were expressed as a percent of volume on day 0, then log transformed, with the *y*-axis representing the logarithm of percent initial tumor volume, where values above the dotted line are greater than the log of 100 (4.60517). The *x*-axis represents days post treatment. In both (**a**,**b**), the red line represents the reconstitution control (RC) group, the green line represents the TPV/Δ2L/Δ66R/FliC group, the blue line represents the TPV/Δ66R/mIL-2 group, and the purple line represents the wtTPV group. In (**a**), the mean log percent of initial tumor volume for TPV/Δ66R/mIL-2 and wtTPV treatments was significantly less than RC treated animals (*p* = 0.01550 * and *p* = 0.04921 *, respectively), and TPV/Δ2L/Δ66R/FliC treatment was not significantly less than RC treatment (*p* = 0.43224). In (**b**), the mean log percent of initial tumor volume for both TPV/Δ2L/Δ66R/FliC and TPV/Δ66R/mIL-2 was significantly less when compared to the mean of wtTPV and RC treatments (*p* = 5.78 × 10^−6^ ***). When assessed individually, the mean log percent of initial tumor volume for TPV/Δ2L/Δ66R/FliC treatment was significantly less than RC (*p* = 0.005195 **), and though less, TPV/Δ66R/mIL-2 treatment was not significantly different than RC (*p* = 0.087323). The log percent of initial tumor volume for RC treatment was also significantly less than wtTPV (*p* = 0.03992 *) in non-injected tumors. * *p* < 0.05, ** *p* < 0.01, *** *p* < 0.001, not significant (NS), *n* = 5–6 mice per group.

**Figure 3 genes-14-01533-f003:**
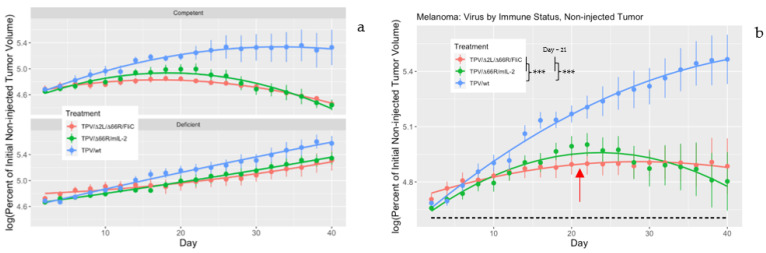
Tumor analysis of TPV-mediated treatments by immune status for directly injected and non-injected tumors. Mice were injected subcutaneously with 5 × 10^6^ SK-MEL3 cells on both flanks and allowed to grow until one tumor reached 120–180 mm^3^ in volume. Once one tumor reached this threshold, it was injected directly with either 5 × 10^6^ PFU/100 μL of TPV or sterile buffer for a mock injection (day 0). On day 4, all mice in the reconstitution arm were injected intraperitoneally with 3 × 10^6^ splenocytes from normal BALB/c donor mice. Both directly injected and non-injected tumor volumes were measured every 2 days for a period of 40 days total. Measurements were taken with Vernier calipers with at least 2 independent measurements per day. Volumes were calculated using the formula: ((length × width × height) × (π/6)). All tumor volumes were expressed as a percent of volume on day 0, then log transformed, with the *y*-axis representing the logarithm of percent initial tumor volume, where values above the dotted line are greater than the log of 100 (4.60517). The *x*-axis represents days post treatment. Factorial ANOVA compared the mean log percent of initial tumor volume for both IDt and ICt animals pooled together for each respective TPV for the directly injected tumors and non-injected tumors. In (**a**,**b**), the red line represents TPV/Δ2L/Δ66R/FliC-treated mice, the green line represents TPV/Δ66R/mIL-2-treated mice, and the blue line represents wtTPV-treated mice. There were no significant differences between any TPV treatment for directly injected tumors when pooled over both immune status types. In (**a**), TPV treatments were separated by immune status for non-injected tumors. Immune-deficient mice had significantly greater mean log percent of initial tumor volume (18%) than splenocyte-reconstituted mice treated with TPV recombinants. In (**b**), the mean log percent of initial tumor volume for wtTPV-treated mice was significantly greater than the mean of both TPV recombinant treatments (*p* = 6.172 × 10^−5^ ***). Midpoint analysis on day 21 (red arrow) also demonstrates significantly greater slopes of mean log percent of initial tumor volume for wtTPV treatment compared to the mean slopes of both TPV recombinants in non-injected tumors (*p* = 2.482 × 10^−5^ ***). *** *p* < 0.001, *n* = 5–6 mice per group.

**Figure 4 genes-14-01533-f004:**
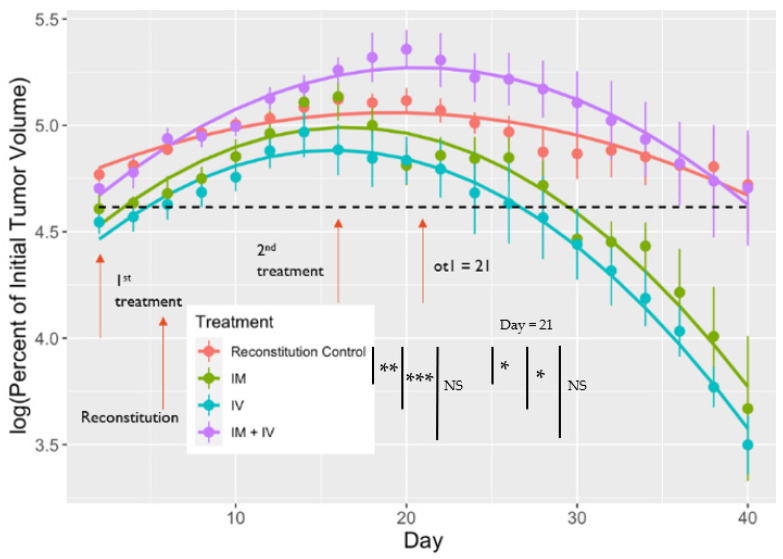
Intravenous and intramuscular administration of TPV/Δ66R/mIL-2 led to significantly faster tumor regression of SK-MEL3 tumor xenografts in BALB/c nude mice reconstituted with splenocytes from normal BALB/c donors. Mice were injected subcutaneously with 5 × 10^6^ SK-MEL3 cells and allowed to grow until the tumor reached 100 mm^3^ in volume. Once the tumor reached this threshold, mice were randomly assigned to 3 treatment groups, either receiving IV, IM, or both IV and IM administration of 1 × 10^6^ PFU/50 μL TPV/Δ66R/mIL-2 (day 0). On day 4, all mice were injected intraperitoneally with 3 × 10^6^ splenocytes from normal BALB/c donor mice. On day 14, mice received a second administration of 1 × 10^6^ PFU/50 µL TPV/Δ66R/mIL-2 in the same way as day 0. Tumor volumes were measured every 2 days for a period of 40 days total. Measurements were taken with Vernier calipers with at least 2 independent measurements per day. Volumes were calculated using the formula: ((length × width × height) × (π/6)). All tumor volumes were expressed as a percent of volume on day 0, then log transformed, with the *y*-axis representing the logarithm of percent initial tumor volume, where values above the dotted line are greater than the log of 100 (4.60517). The *x*-axis represents days post treatment. The red line represents reconstitution control treatment, the green line represents IM administration of TPV/Δ66R/mIL-2, the blue line represents IV administration of TPV/Δ66R/mIL-2, and the purple line represents simultaneous administration of TPV/Δ66R/mIL-2 both IV and IM. When compared to RC, IV administration and IM administration of TPV/Δ66R/mIL-2 both significantly regressed tumor volumes over the course of the experiment (*p* = 1.57 × 10^−5^ *** and *p* = 0.00278 **, respectively). Analysis of tumor growth curve slope trajectory at experimental midpoint (day 21) showed that IV administration of TPV/Δ66R/mIL-2 was significantly more negative than RC treatment (*p* = 0.03563 *). Analysis of the shape of each growth curve at the experimental midpoint showed that both IV and IM administration of TPV/Δ66R/mIL-2 were significantly more negative than RC treatment (*p* = 0.008985 ** and *p* = 0.010054 *, respectively). * *p* < 0.05, ** *p* < 0.01, *** *p* < 0.001, not significant (NS), *n* = 4 for each TPV group, *n* = 6 for the RC group.

**Table 1 genes-14-01533-t001:** Endpoint analysis of tumor volume for directly injected tumors. Holm adjustment method.

Group Comparison	Value	df	Sum of Square	F Value	*p*-Value of F
Control-TPV/Δ2L/Δ66R/FliC	4.2654	1	103.96	9.9686	0.01635 *
Control-TPV/Δ66R/mIL-2	4.7913	1	117.23	11.2405	0.01159 *
Control-wtTPV	2.6760	1	37.36	3.5824	0.26675
TPV/Δ2L/Δ66R/FliC-TPV/Δ66R/mIL-2	0.5260	1	1.48	0.1415	0.70910
TPV/Δ2L/Δ66R/FliC-wtTPV	−1.5894	1	13.78	1.3213	0.51632
TPV/Δ66R/mIL-2-wtTPV	−2.1153	1	21.92	2.1016	0.46815

* *p* < 0.05.

## Data Availability

Not applicable.

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
