# Peer review of "Multiple Administration Routes, Including Intramuscular Injection, of Oncolytic Tanapoxvirus Variants Significantly Regress Human Melanoma Xenografts in BALB/c Nude Mice Reconstituted with Splenocytes from Normal BALB/c Donors"

_genes, 2023, doi:10.3390/genes14081533_

Round 1

Reviewer 1 Report

This is a compelling, novel study assessing the utility of Tanapox virotherapy in a melanoma xenograft model. Given the viral tropsim constraints the authors attempt to overcome immunocompetent deficiencies via adoptive transfer of splenocytes from naive hosts. This provides compelling antitumor responses following local and systemic viral deliver. The authors provide no evidence that IV/IM viral delivery is associated with tumor localization and/or infection. Nor is any evidence provided demonstrating enhanced TIL (vs control) in virus delivered cohorts following adoptive transfer. In addition, it is hard to make a case for systemic Tanapox delivery based on this limited model as being translatable to humans given this reconstitution model lacks circulating antibodies and other effectors that rapidly deplete virions (especially large enveloped viruses) present in immunocompetent hosts. No data demonstrating serum or tumor titers of virus following delivery were presented. There is no evidence of viral transgene expression provided nor the consequential host TLR/IL-2 response. Despite these mechanistic oversights, overall given the constraints of this virus in immunocompetent modeling this is a promising report. 

Author Response

This is a compelling, novel study assessing the utility of Tanapox virotherapy in a melanoma xenograft model. Given the viral tropsim constraints the authors attempt to overcome immunocompetent deficiencies via adoptive transfer of splenocytes from naive hosts. This provides compelling antitumor responses following local and systemic viral deliver. The authors provide no evidence that IV/IM viral delivery is associated with tumor localization and/or infection. Nor is any evidence provided demonstrating enhanced TIL (vs control) in virus delivered cohorts following adoptive transfer. In addition, it is hard to make a case for systemic Tanapox delivery based on this limited model as being translatable to humans given this reconstitution model lacks circulating antibodies and other effectors that rapidly deplete virions (especially large enveloped viruses) present in immunocompetent hosts. No data demonstrating serum or tumor titers of virus following delivery were presented. There is no evidence of viral transgene expression provided nor the consequential host TLR/IL-2 response. Despite these mechanistic oversights, overall given the constraints of this virus in immunocompetent modeling this is a promising report. 

Response:

Thank you very much for reviewing our manuscript. To provide clarity in regard to some of your comments in the above paragraph we would like to address a few things. First, our study was designed around the development of this tumor model and analyses of TPV’s efficacy against melanoma tumors using growth curve modeling. We would have liked to run additional experiments looking at tumor pathology in treated and non-treated tumors and the infiltrating immune cells responding to our viruses as well as cytokine analyses demonstrating the in vivo consequences of treatment. Unfortunately, some of the proposed experiments in your comments were beyond the scope of this study. For your comment regarding presence of effectors that would rapidly deplete virions, while it is true that these nude mice under normal conditions would be limited to IgM antibody production with little to no effect on TPV’s circulation, BALB/c nude mice, and other nude mice breeds, have compensatory macrophage and natural killer cell activity and slightly larger populations for their lack of mature T cell functionality. Both of these cell types are capable of virion clearance through phagocytosis. This is all prior to adoptive transfer which would supplement the missing mature T cell population and allow for the potential production of neutralizing antibodies roughly 2 weeks later as active TPV infection of tumor cells would be ongoing. We have demonstrated viral transgene expression in nude mice previously in the treatment of these same tumor cells (reference 3) and FliC expression was demonstrated in another reference (5). We added clarification on lines (476-477 and 478-480) in the text. We have also added an entire paragraph at the end of the discussion addressing limitations associated with the study.

Reviewer 2 Report

Thank you for allowing me to review this original article entitled "IT, IV, and IM administration of Oncolytic Tanapoxvirus Variants Significantly Regress Human Melanoma Xenografts in BALB/c Nude Mice Reconstituted with Splenocytes from Normal BALB/c Donors".

The topic is intriguing, the study is well described, and the in vivo experiments show Tanapoxvirus tested for efficacy in a new xenograft mouse model.

I have just some minor observations: 

-I suggest avoiding abbreviations in the manuscript title (for example, IT, IV and IM)

-In the introduction section, line 52, I suggest replacing the word "melanoma " with "cutaneous melanoma."

- The result section could be more precise. Some points need to be clarified; in the manuscript, some correct comments are added as a note (please see "SK comments"), but I need to know if they belong to the authors.

Author Response

Thank you for allowing me to review this original article entitled "IT, IV, and IM administration of Oncolytic Tanapoxvirus Variants Significantly Regress Human Melanoma Xenografts in BALB/c Nude Mice Reconstituted with Splenocytes from Normal BALB/c Donors".

The topic is intriguing, the study is well described, and the in vivo experiments show Tanapoxvirus tested for efficacy in a new xenograft mouse model.

I have just some minor observations: 

-I suggest avoiding abbreviations in the manuscript title (for example, IT, IV and IM)

We basically agree with the reviewer’s comment here. Since we would like to emphasize an additional potent way of injecting oncolytic virus in this model, we thought the acronyms would grab the attention of the readers. However, based on this feedback we have instead decided to modify the title to, “Multiple Administration Routes, Including Intramuscular Injection, of Oncolytic Tanapoxvirus Variants Significantly Regress Human Melanoma Xenografts in BALB/c Nude Mice Reconstituted with Splenocytes from Normal BALB/c Donors”.

-In the introduction section, line 52, I suggest replacing the word "melanoma " with "cutaneous melanoma."

We have replaced “melanoma” with “cutaneous melanoma” on line 52.

- The result section could be more precise. Some points need to be clarified; in the manuscript, some correct comments are added as a note (please see "SK comments"), but I need to know if they belong to the authors.

The comments on the side bar were resolved comments from one of the authors, Steve Kohler, in a draft prior to submission, but were not properly deleted from the Word document. Though they were hidden, they appeared when the document was converted to a pdf. These are deleted from the corrected manuscript.
